# How Healthy Are Health-Related Behaviors in University Students: The HOLISTic Study

**DOI:** 10.3390/nu13020675

**Published:** 2021-02-19

**Authors:** Hellas Cena, Debora Porri, Rachele De Giuseppe, Aliki Kalmpourtzidou, Fiorella Pia Salvatore, Marwan El Ghoch, Leila Itani, Dima Kreidieh, Anna Brytek-Matera, Cristina Bianca Pocol, Donaldo Segundo Arteta Arteta, Gözde Utan, Ivana Kolčić

**Affiliations:** 1Laboratory of Dietetics and Clinical Nutrition, Department of Public Health, Experimental and Forensic Medicine, University of Pavia, 27100 Pavia, Italy; hellas.cena@unipv.it (H.C.); debora.porri01@universitadipavia.it (D.P.); alikikalb@hotmail.com (A.K.); goezde.utan01@universitadipavia.it (G.U.); 2Clinical Nutrition and Dietetics Service, Unit of Internal Medicine and Endocrinology, ICS Maugeri IRCCS, 27100 Pavia, Italy; 3Department of Economics, University of Foggia, 71121 Foggia, Italy; fiorellapia.salvatore@unifg.it; 4Department of Nutrition and Dietetics, Faculty of Health Sciences, Beirut Arab University, P.O. Box 11-5020 Riad El Solh, Beirut 11072809, Lebanon; m.ghoch@bau.edu.lb (M.E.G.); l.itani@bau.edu.lb (L.I.); d.kraydeyeh@bau.edu.lb (D.K.); 5Institute of Psychology, University of Wroclaw, 50-527 Wroclaw, Poland; anna.brytek-matera@uwr.edu.pl; 6Department of Animal Production and Food Safety, University of Agricultural Sciences and Veterinary Medicine of Cluj Napoca, 400372 Cluj Napoca, Romania; cristina.pocol@usamvcluj.ro; 7Department of Physiology, Anatomy and Cell Biology, Universidad Pablo de Olavide, 41013 Seville, Spain; dsartart@upo.es; 8Department of Public Health, University of Split School of Medicine, 21000 Split, Croatia; ikolcic@mefst.hr

**Keywords:** university students, lifestyle habits, lifestyle medicine, health, public health, prevention, NCDs

## Abstract

The aim of this cross-sectional study was to assess the health-related behaviors among university students, with emphasis on health sciences students from Croatia, Italy, Lebanon, Poland, Romania, Spain and Turkey. We included 6222 students in Medicine, Dentistry, Nursing, Pharmacy, Nutrition and Dietetics, Sports Sciences, Veterinary, and Economics enrolled between April 2018 and March 2020. We assessed dietary patterns, sleeping habits, physical activity and perceived stress among students by means of validated questionnaires. The median age ranged between 19 and 24 years, smoking prevalence between 12.0% and 35.4%, and body mass index (BMI) ranged between 21.1 and 23.2 kg/m^2^. Breakfast was less often and more often consumed daily in Turkey (36.7%), and Italy (75.7%), respectively. The highest Mediterranean diet score was recorded in Spain and Italy, and the lowest in Turkey, followed by students from Croatia, Lebanon, Poland and Romania. Sleep duration, physical activity and stress perception also differed between countries. Multivariable regression analysis revealed a small, but positive association between BMI and several characteristics, including age, female gender, smoking, physical activity, mobile phone use, and perceived stress. A negative association was found between BMI and sleep duration on non-working days. Self-rated health perception was positively associated with female gender, breakfast, physical activity, and time spent studying, and negatively with BMI, smoking and stress. Our results demonstrated diverse habits in students from different countries, some of which were less healthy than anticipated, given their educational background. Greater emphasis needs to be placed on improving the lifestyle of these adolescents and young adults, who will be tomorrow’s healthcare workers.

## 1. Introduction

Adolescence and young adulthood (10–24 years of age) are usually thought to be the healthiest stages of life [1], but young people face many challenges in becoming adults in our modern and rapidly shifting society. These challenges include, but are not limited to, social inequalities, diseases due to poverty, injuries and violence, mental health problems, substance use disorders, and adolescence-emerging behavioral health risks for the development of non-communicable diseases (NCDs) later in life, such as tobacco and alcohol use, physical inactivity, and poor diet [1].

Considering the extensive NCD global burden [2], prevention strategies should focus on decreasing exposure to major lifestyle-related risk factors that develop early in life. Adolescence, as a transitional phase of life, provides a unique opportunity with promises of fostering and strengthening future health and wellbeing [3]. This is particularly important nowadays, when for the first time in history, the adolescent population in the world is the largest in history, close to 1.8 billion [4]. Today these adolescents smoke less, but they gain excessive weight and are frequently exposed to high and early alcohol consumption [4]. Hence, extending a global strategy for adolescence health is mandatory. About 250 million more teenagers and young people are now exposed to these threats compared to 25 years ago [4], which is capable of reshaping the landscape of future life expectancy. A recent study showed that US life expectancy plateaued in 2011 and has actually started to decrease after 2014 [5]. These findings were attributed to increased specific mortality due to drug overdoses, alcohol abuse, suicides, cardiovascular diseases and diabetes in young and middle-aged adults [5]. Lifestyle-related risk factors are notably behind these trends, especially in the obesity epidemic, as well as in mood disorders, which have increased in US adolescents and young adults [5].

Furthermore, psychological stress and uncontrollable highly stressful events were found to be associated with substance addiction, adiposity and weight gain, as well as with unhealthy dietary patterns such as skipping meals, restraining intake or, on the contrary, binge eating [6]. Stress has also been associated with increased consumption of fast food, snacks, and energy-dense and highly palatable foods [6,7]. Additionally, a negative association between stress and diet quality was confirmed among 12–17 years old adolescents from Europe [8]. Evidently, different aspects of lifestyle and health-related behaviors are closely intertwined. Psychological wellbeing is strongly connected to physical health, while our habits and daily lives are immersed into this 24-h society where time constraints are no longer “restricting” the everyday life [9]. Unfortunately, we are victims of this vicious circle with perpetual health erosion due to our modern lifestyle. At the same time, besides an increase in unhealthy lifestyles, personal responsibility for health (“new health consciousness”), known as “healthism” [10], is gaining ground, enabling individuals to gain awareness and behaviors, including regular exercise and healthy eating, aimed at maintaining health and avoiding disease [11]. However, a growing trend and dangerous sliding towards disturbed behaviors, such as excessive concern about diet and health with miraculous expectations is observed, contributing to the development of persistent fixation about healthy eating, as in orthorexia nervosa (ON) [12]. 

Our ultimate goal should be morbidity and mortality decrease and wellbeing increase across populations of all age groups with an appropriate start during education and academic training, focusing in particular on healthcare students, who will have to acquire an adequate mastery for their future role of healthcare workers. Is there any better start than the academic training of those who will take care of others’ health in the future? Unfortunately, the medical education system is largely failing to equip students with knowledge about nutrition [13,14], physical activity [15] or stress management, pillars of a healthy lifestyle. This will bridge the substantial gap between the actual demands of patients and the inability to meet them by the healthcare system for effective NCD prevention and treatment.

In particular, undergraduate healthcare students may have a low quality of life and experience greater difficulties due to hospital shifts, high study load, stress and general burnout [16]. Therefore, healthcare students are at risk of adopting an unhealthy lifestyle, including excessive alcohol consumption, smoking, and unhealthy dietary patterns, with the overconsumption of comfort foods, which are commonly used as copying mechanisms with chronic psychological distress [17]. These same students, who will become future health professionals, need, more than others, to be aware of healthy lifestyle benefits and present themselves as healthy role models to promote a healthy lifestyle in the population and build resilience to mental stress [18].

Hence, the primary objective of the HOLISTic study (habits, orthorexia nervosa and lifestyle in university students) is to assess health-related behaviors and lifestyle in university students of different cultural and educational backgrounds. Specifically, we aimed to provide an insight into the overall dietary patterns, sleeping habits, physical activity and perceived stress among students, which will enable us to design tailored educational interventions to improve health and wellbeing in this group of adolescents and young adults. In addition, healthcare students must set an example for their future patients, acquiring knowledge, confidence and competence to deliver in turn health interventions and healthy lifestyle advices.

## 2. Materials and Methods

The present multicenter cross-sectional study has been carried out in eight universities from seven European and non-European countries, including Croatia, Italy, Lebanon, Poland, Romania, Spain and Turkey. We have included undergraduate students of different faculties, placing special emphasis on health sciences, as presented in Appendix A. Students in Medicine, Dentistry, Nursing, Pharmacy, Dietetics, Sports Sciences, Veterinary Sciences and Economics were enrolled, and no exclusion criteria were applied. We obtained high response rates in most of the study sites, except for some of the study years in Lebanon, Pavia in Italy, and Romania. The response rate varied between 69.6% and 97.1% in Croatia, 62.4% and 100.0% in Foggia, 4.3% and 88.2% in Pavia, 11.3% and 96.0% in Lebanon, 84% and 92% in Poland, 5% and 81% in Romania, 36% and 100% in Spain, and 52% and 92% in Turkey (Appendix A).

The study was conducted in agreement with the national and international regulations on biomedical research and the Declaration of Helsinki. Ethical Committee approvals were obtained for each country, where applicable according to the institutional policies (Appendix A). There were no direct benefits to the respondents from participating in this study. Participants were fully instructed about the study aim and requirements and were also informed that by agreeing to fill in the questionnaire they confirmed their participation, automatically providing an informed consent in all universities, except in Koç University in Turkey, where students effectively signed an informed consent before completing the questionnaire. Based on the study site preferences, the anonymous questionnaire was delivered either as a paper-based or online survey. For the paper-based survey, names and surnames of the subjects were not collected and all the subjects were assigned a unique code, thus ensuring anonymity and privacy protection. The anonymous online survey was administered by means of Google forms and data for each study site and was downloaded as a Microsoft Excel sheet (Appendix A).

### 2.1. Data Collection

The collection of the data, performed by using either online or paper-based anonymous surveys, occurred between April 2018 and March 2020 (Appendix A). In both cases, the maximum time expected to complete the survey was between 20 and 30 min. The survey consisted of seven sections which aimed at obtaining: (i) general information; (ii) dietary habits; (iii) eating behavior; (iv) lifestyle and nutrition knowledge; (v) physical activity; (vi) sleeping habits and screen-time, and (vii) perceived level of stress and quality of life. 

Each section was explored by either general multiple-choice or open-ended questions or previously validated questionnaires, as presented in Appendix A. Where the language differed from the native language of the instrument (e.g., English), the questionnaire was adapted by using a forward–back translation (FBT), as stated in Appendix A. Particularly, the aim of this process was to achieve different language versions of the native language instrument, conceptually equivalent in each of the target countries/cultures. Therefore, the implementation method consisted of: (i) forward translation; (ii) expert panel back translation; (iii) pre-testing and cognitive interviewing; (iv) final version, as per instructions for adaptation of instruments described elsewhere [19].

(i)General information

Gender and age were recorded together with the type of study program, year of attendance and grade point average (GPA), throughout the program of study, recoded into percentiles since different universities had different scoring schemes, where the maximum possible grades represented 100%, and the minimum for passing was set at 0% in each country. 

Self-reported weight (Kg) and height (cm) were recorded, as well as date of the last body weight measurement. Body mass index (BMI) was calculated using the standard formula ((BMI = weight (Kg)/height (m^2^)).

Students were classified as smokers, non-smokers or former smokers. Students were also asked about their self-perceived health, using a Likert scale, with values ranging between zero and 10, where zero meant very sick, and 10 represented full health. 

(ii)Dietary habits

Dietary habits, including food preferences, breakfast frequency during the week, daily number of main meals and snacks (recorded separately for working days and off-days), lunch and dinner time, snacking frequency while studying or TV watching, and type of snacks consumed were recorded through multiple choice or open-ended questions.

Using the Mediterranean Diet Serving Score (MDSS) system, which is based on the latest update of the Mediterranean diet pyramid [20], we assessed the Mediterranean diet (MD) adherence. The MDSS takes into account the recommended frequency of consumption of different foods and food groups. Individuals who had intakes in line with the number of recommended servings per day or week were awarded with 3, 2, or 1 point for each of the food groups [20]. This approach places greater emphasis on the foods that should be consumed at each meal (fruit, vegetables, olive oil, grains), followed by those that should be consumed daily (dairy and dried fruit and nuts), and finally, those that should be consumed weekly (potatoes, legumes, eggs, fish, white meats, red meats, sweets) [20]. For alcohol intake, 1 point for 1 or 2 glasses of wine (fermented drinks) were added for women and men, respectively [20]. The minimum and maximum values of this scoring system are, respectively, zero and 24, where higher scores indicate greater MD compliance, with a cut-off point of 14 indicating good MD adherence [20].

(iii)Lifestyle and nutrition knowledge

Self-perceived desire to learn more about the association between lifestyle and health was ranked on a 5-point scale, where 1 corresponded to “not necessary, I know everything about it” and 5 to “I would like very much to learn more”. 

(iv)Orthorexic eating behavior

Eating behavior was assessed by using both the Eating Habits Questionnaire (EHQ) [21] and orthorexia nervosa test (ORTO-15) [22]. The ORTO-15 test investigates the obsessive attitude of the individuals in choosing, buying, preparing and consuming food considered healthy (e.g., “Are you willing to spend more money to have healthier food?”, “Do you think your mood affects your eating behavior?”) [22]. The test has a minimum score of 15 and a maximum of 60, where a lower score corresponds to higher orthorexic symptoms, and score <40 has been proposed as cut-off value correspondent to greater probability of being orthorexic [22]. As for the questionnaires used, Italy, Lebanon, Turkey, Spain and Poland adopted their country-specific validated/adapted national versions [22,23,24,25].

(v)Physical activity

Physical activity during the last 7 days was investigated by using the validated International Physical Activity Questionnaire, short form (IPAQ; short form) [26]. Metabolic equivalent of task (MET-min) per week were calculated as:

METs = MET level * minutes of activity * events per week 

All study sites downloaded the appropriate adapted country/language specific IPAQ version [27], except Lebanon, Romania and Turkey that used the English version one. 

(vi)Sleeping habits and screen-time

Sleeping habits included information on the usual bedtime at night during working days and separately on weekends or holidays, and awaking time, using open-ended questions. With these data, we calculated the sleeping time on working days and on non-working days. 

Screen time was recorded as the average daily time (hours) students spent watching TV, using computers/tablets and mobile phones (separate open-ended questions). Additionally, students were asked about the average time spent studying daily.

(vii)Perceived level of stress

Perceived level of stress was explored by using the Perceived Stress Scale-10 (PSS-10) [28]. The questionnaire evaluates the degree to which external demands appear to be higher than the individual’s perceived capability to handle the situation. The scale has a minimum score of 0 and a maximum of 40, where a higher score indicates higher perceived stress during last month. Particularly, Spain adopted the PSS-14 validated/adapted country-specific version [29], which was converted to PSS-10 to make data comparable between study sites.

### 2.2. Statistical Analysis 

Data were reported as absolute numbers and percentages for categorical variables, while medians and interquartile ranges (IQR) were used to describe ordinal variables and numerical variables, which did not follow normal distribution (tested with the Kolmogorov–Smirnov test). We used the chi-square test for categorical variables, and the Kruskal–Wallis test (with the Mann–Whitney U test as post-hoc test) and Spearman’s rank correlation test for ordinal and numerical variables in bivariate analysis. Additionally, we created two multivariable linear regression models, with BMI and self-rated health perception as outcome variables. Predictor variables simultaneously entered in the model included age, gender, smoking, BMI (only for self-perceived health model), cohort (study site), MDSS score, breakfast frequency, sleep duration on working days and non-working days, total METs per week, mobile phone use time, study time, PSS-10 score and self-rated health perception (only for BMI model). Both models had a good model fit, expressed as the Durbin–Watson test, which was 1.951 for the BMI model (adjusted R^2^ = 15.2%) and 1.857 for the self-perceived health model (adjusted R^2^ = 11.4%).

Missing data were handled by excluding subjects with missing data from the analysis on a case-by-case basis (we did not delete entire observations with missing data in order to prevent wasting of collected data). Besides some differences in questionnaire between study sites (Appendix A), missing data per variable ranged between 0.03% and 0.98%.

Statistical analysis was performed using IBM SPSS Statistics software (v21.0; IBM, Armonk, NY, USA), with statistical significance set at *p* < 0.05.

## 3. Results

The total number of students enrolled in the study was 6222. The sample consisted mainly of healthcare students (*n* = 4608, 74.1%). The median age ranged from 19 to 24 years old (Table 1). The majority of students were females, except in Spain, where females represented about 2/5 of all the students enrolled (39.8%). 

The highest percentage of current smokers was observed in Romania (35.4%), and the lowest in Spain (12.0%). BMI ranged from 21.1 kg/m^2^ to 23.2 kg/m^2^, corresponding, respectively, to the median BMI reported in northern Italy (University of Pavia) and in Lebanon, as shown in Table 1, while post-hoc analysis of differences between countries is presented in Appendix A. 

The self-rated health perception was rather high (median eight out of 10 in all countries, except in Croatia, where it was nine out of 10), with statistically significant differences between in most of the countries, except between students from Foggia and Romania, Pavia and Romania, Lebanon and Poland, Lebanon and Turkey, and Poland and Turkey (Table 1 and Appendix A). 

The median MDSS score was below the cut-off value of 14 in all study sites, ranging from five in Turkey to 10 in northern Italy and Spain. Students from Croatia and from Lebanon scored the same MDSS median as students from Poland, lower than students from Romania, while students from Turkey obtained the lowest median MDSS score (Table 1). Both in southern (University of Foggia) and northern Italy (University of Pavia), students consumed breakfast more frequently on a daily basis (75.7% and 75.3%, respectively), compared to Turkey, where only 36.7% of students consumed breakfast daily. 

In all countries, the median ORTO-15 score was slightly below 40; the lowest was recorded in students from Foggia (35.0, interquartile range (IQR) 6.0), and the highest in students from Poland (38.0 (8.0)), with statistically significant differences between most of the countries (Table 1 and Appendix A). 

The median sleep duration during working days ranged from 7.0 to 8.0 h, and on non-working days medians ranged from 8.0 to 9.0 h. Comparison between countries revealed that students from Croatia, Lebanon and Turkey reported the lowest median sleep duration on working days, while students from Italy, Poland and Spain reported longer sleep duration (Table 1 and Appendix A). Students with shorter sleep duration during working days had longer sleep duration on non-working days.

Students from Spain reported the highest physical activity estimated by total METs per week, while students from Turkey reported the lowest. Lebanese students tended to use their mobile phone for longer time during the day compared to all other students. Students from Poland, Romania and Spain spent the least amount of time studying per day, while students from southern Italy reported the highest daily time studying (median of 5 h (IQR 3.0)). 

According to the PSS-10 score, students’ stress perception was moderate in all countries, with the lowest score recorded in Spain (Table 1). Students were interested in learning about the relationship between lifestyle and health outcomes, with 49.3% of students from Croatia explicitly asking for more education on this topic, similar to what was reported by students from Lebanon (48.9%), and Spain (48.8%) (Table 1).

The correlation analysis including the entire sample of students revealed a relatively low, but significant correlation between the MDSS and breakfast frequency during the week (*r* = 0.22; *p* < 0.001) (Table 2). Additionally, the MDSS was correlated to sleep duration on working days (*r* = 0.10; *p* < 0.001), total METs per week (*r* = 0.16; *p* < 0.001), study time (*r* = 0.13; *p* < 0.001), age (*r* = 0.10; *p* < 0.001), and a weak correlation was found between the MDSS and self-rated health perception (*r* = 0.03; *p* = 0.008). A negative correlation was recorded between the MDSS score and ORTO-15 score (*r* = −0.15; *p* < 0.001), and sleep duration on non-working days (*r* = −0.10; *p* < 0.001). However, we did not find any correlation between the MDSS score and BMI, mobile phone use or stress score (Table 2). BMI was correlated with breakfast frequency (*r* = −0.03; *p* = 0.009), ORTO-15 (*r* = −0.09; *p* < 0.001), sleep duration during non-working days (*r* = −0.05; *p* < 0.001), total METs (*r* = 0.09; *p* < 0.001), mobile phone use (r = 0.04; *p* = 0.003), self-rated health perception (*r* = −0.05; *p* < 0.001), and age (*r* = 0.14; *p* < 0.001) (Table 2).

Regression analysis confirmed some of these results (Table 3). For example, we found positive associations between BMI and age (*β* = 0.13; *p* < 0.001), female gender (*β* = 0.30; *p* < 0.001), active smoking (*β* = 0.03; *p* = 0.016), total METs (*β* = 0.04; *p* = 0.002), mobile phone use (*β* = 0.05; *p* < 0.001), perceived stress score (*β* = 0.04; *p* = 0.002), and a negative association between BMI and sleep duration during non-working days (*β* = −0.03; *p* = 0.040) and self-rated health perception (*β* = −0.10; *p* < 0.001). Self-perceived health was associated with female gender (*β* = 0.03; *p* = 0.018), active smoking status (*β* = −0.06; *p* < 0.001), breakfast frequency (*β* = 0.04; *p* = 0.003), total METs (*β* = 0.08; *p* < 0.001), study time (*β* = 0.03; *p* = 0.039), and perceived stress (*β* = −0.23; *p* < 0.001). Compared to students from Croatia, students from all other study sites had lower self-rated health perception, and students from Pavia (*β* = −0.10; *p* < 0.001), and Poland (*β* = −0.03; *p* = 0.009) had lower BMI compared to students from Croatia, while students from Lebanon had higher BMI (*β* = 0.13; *p* < 0.001) (Table 3). 

Mediterranean diet adherence was not associated with either BMI or self-rated health perception, the same as sleep duration during working days (Table 3).

## 4. Discussion

The preliminary description of our multicenter study including 6222 university students shows an interesting pattern of health-related behaviors across different countries. Students in some countries showed unhealthier habits grouped together, such as higher prevalence of smoking, skipping breakfast and shorter sleep duration on working days, as recorded in Romania, Lebanon, Turkey and in Croatia. On the other hand, students from Italy and Spain displayed higher adherence to the MD, higher daily breakfast consumption, they smoked less, and slept longer on working days. Additionally, we found that lifestyle was associated with health outcomes, such as BMI and self-perceived health, even in these young and generally healthy university students. 

In our sub-samples, BMI varied within physiological values, although recent studies on young Europeans showed increasing trends in overweight and obesity [30,31]. However, students from Pavia in Italy and students from Poland had a lower BMI compared to students from Croatia, while students from Lebanon had the highest BMI. We found a positive association between BMI and age, BMI and female gender, as well as BMI and smoking habits. Finding a higher BMI in active smokers is contradictory to previous evidence reported in adolescents [32]. However, studies show that those who are more nicotine dependent are more likely to gain weight over time, and pack-years are associated to greater waist circumference, higher visceral adipose depot and higher risk of metabolic syndrome [33,34].

Interestingly, BMI was also positively associated with self-reported total physical activity, expressed as total METs per week. This positive association of self-reported physical activity with BMI could be due to the greater muscle mass, as BMI is a simple weight indicator for height and does not reflect body composition. Moreover, median METs per week across all countries indicated moderate/high physical activity level in the whole sample. Still, although it has been previously reported that sports practice, performed also by overweight subjects, contributes to less sedentary activity, nevertheless it has not been correlated to healthy eating habits [35]. On the other hand, since studies generally indicate an inverse relationship between BMI and physical activity levels [36], a more plausible explanation might be that students with higher BMI, although within the normal range, were more engaged in sports practice. Furthermore, in support of this latter speculation, only a weak association between physical activity and BMI in non-obese subjects has been previously described [36].

BMI was also positively associated with mobile phone use and perceived stress level, both of which were positively correlated, though weakly. This association is partly supported by reports from a recent systematic review and meta-analysis showing that college students with mobile phone addiction were more likely to develop increased anxiety, depression, and impulsivity levels and are more likely to suffer from poor sleep quality [37]. In fact, evidence has shown that the association between mobile phone addiction and several psychological and behavioral issues, including stress, anxiety and depression, might be mediated by interpersonal problems [37]. Based on the interpersonal theory, individuals with mobile phone addiction usually neglect the real-world social networking, resulting in reduced social support resources, and higher levels of anxiety and depression [37]. On the other hand, there is also evidence that psychopathology per se may cause mobile phone addiction since mobile devices are often used as a coping strategy in people with anxiety and depression to eliminate their negative emotions [37].

It is well known that sleep duration is an important regulator of body weight and metabolism and the relationship between shorter habitual sleep time and BMI increase has been extensively studied in large population samples [38]. As previously reported, university students are vulnerable to sleep problems, including irregular sleep schedules and sleep deprivation [39]. Indeed, many students go to sleep late and wake up early to attend classes and start their daily life without achieving adequate sleep [39]. Thus, they extend sleep on non-working days (e.g., weekends) to compensate for suboptimal sleep duration on workdays [39]. This was indeed recorded in most of our sample population (with the exception of Foggia) who reported a shorter sleep duration during working days than non-working days. Hence, students with the shortest sleep duration on working days (Croatia, Lebanon and Turkey) reported the longest sleep extension on non-working days. Although this practice may provide temporary relief, overcompensation worsens the problem, leading to further disruption in sleep-wake cycles [39]. This irregular sleep-wake cycle is related to changes in metabolic hormones (e.g., leptin and ghrelin) with consequent increased energy intake and weight gain [38]. Additionally, evidence suggests that increased energy intake is also related to short sleep duration, which increases motivation to seek out food as reward [40]. Indeed, short sleep duration has been acknowledged as a predictor for obesity [39]. Finally, we have also observed a negative association between BMI and sleep duration during non-working days.

Contrary to our expectation, Mediterranean diet (MD) adherence was associated with neither BMI nor self-perceived health. This interesting finding may be due to the fact that our sample showed low MD adherence according to the MDSS, even in those countries known for their traditional MD patterns, such as Spain and Italy, in agreement with previous findings of low MD adherence and food knowledge among healthcare students from Italy [41], Spain [42], and Turkey [43]. A decline in MD adherence of the general population in the Mediterranean area, and especially of younger generations, is widely recognized [44,45,46]. It has been reported that potential factors resulting in moving away from the MD model include globalization, a transition to Western habits, modernization and changes in lifestyle and environment [46]. This occurs especially in adolescents, who tend to be more exposed and influenced by food-related environmental factors linked to nutritional transition, such as food advertising and promotion, as well as meals eaten away from home and the frequent consumption of comfort energy-dense foods [46]. Unfortunately, our findings confirm this phenomenon among students from the Mediterranean area. For example, students from the Adriatic part of Croatia and students from Lebanon and Turkey scored the same median MDSS as students from Poland, or it was even lower than in students from Romania. 

Concerning orthorexic behaviors, university students from Foggia in Italy, and students from Croatia, Romania, and Turkey reported higher risk of ON. A recent study among Italian, Polish and Spanish students showed similar results, with a high prevalence of ON in more than a third of the whole sample [47]. In ON, the intrusive, food-related thoughts generate emotional consequences, such as severe distress, feelings of guilt, and shame [48]. The avoidance of certain food groups may lead to disordered eating attitudes and behaviors, as well as physiological impairment (e.g., malnutrition and weight loss). A previous study [49] has shown that eating-related variables (pathological eating, eating pattern, and MD), compulsive symptoms, and subjective social status were predictors of ON in adults. It is possible that university students who adopt a Mediterranean eating pattern are more likely to pay attention to healthy eating and are more concerned about food, avoiding all foodstuffs subjectively considered to be “unhealthy” [50]. This is a topic worthy of further exploration. Behavioral patterns that develop during the transition from adolescence to adulthood (18–25 years) often persist later in life, affecting individuals’ health, as well as their partners and/or their children [51], therefore healthy eating habits acquired during youth are important pillars for ensuring future health.

Finally, even though the perceived stress score in our cohort indicated moderate levels of stress, it is well known that university students, especially healthcare ones, may perceive lower quality of life and less compliance to healthy lifestyle due to hospital shifts and high course loads and performance pressure, with a consequent high level of stress [52].

Self-rated health perception was slightly higher in female students, in those who ate breakfast more frequently, in those who reported higher level of physical activity and spent more time studying. This indicates that morning habits, physical exercise and study engagement and sex were important factors influencing the self-perceived health rating of university students included in this study, thus also affecting their future health. On the other hand, self-rated health perception was lower in active smokers, in students with higher BMI, and especially so in students who had higher level of perceived stress. Interestingly, compared to students from Croatia, students from all other study sites had a slightly lower health perception. This warrants further investigation and data analysis within our study. 

Actually, we plan to perform subsequent analyses, aiming at investigating detailed aspects of lifestyle and health-related behaviors in university students of health sciences, comparing dietary habits, and in particular MD components and MD compliance, between Mediterranean and non-Mediterranean countries. Furthermore, we plan to determine the prevalence of ON and to describe orthorexic eating behaviors and their association to other dietary patterns, and other lifestyle variables (e.g., physical activity, smoking habits, sleeping habits, and stress). Additionally, we plan to investigate physical activity patterns in students, sleeping habits in detail, as well as self-estimated levels of perceived stress and quality of life among students from different cultural and educational backgrounds. Ultimately, our goal is to obtain information on the prevalence of unhealthy habits in health sciences students, in order to be able to tailor specific targeted interventions. These would include broad educational approaches and specific workshops to increase students’ healthy lifestyle knowledge and practice, which will result in primary disease prevention in this young and still healthy group of adolescents and young adults. 

Investments to improve nutrition and health focusing on increasing nutrition knowledge as well as raising awareness of the benefits of a healthy lifestyle are crucial during adolescence. Even though “lifestyle’’ can be defined as a “way of living’’, this is an umbrella term that encompasses many different concepts, such as the way an individual lives, works, eats, moves, entertains, sleeps and relaxes, shaped not only by personal habits but also by broader cultural, social, political and economic factors. We have never published so many scientific studies on lifestyle impact on health, yet we are still far from having a complete understanding. Additionally, media continues to broadcast minor and uncertain issues in lifestyle science, creating confusion and diverting public attention from the key challenges, major health problems and largest burden of disease [53]. Unfortunately, even university curricula in health sciences are falling behind in teaching students about nutrition [14,54], physical activity and stress management [15,54]. For instance, human nutrition training should be a part of different health/science degree programs, with specific training programs based on different professional needs [14]. Moreover, students should be educated about food systems and the sustainability of different dietary patterns, emphasizing food quality and sustainability, corresponding to reduced meat and dairy product consumption and increased consumption of fruit and vegetables, essential for both the health of the planet and human health [55,56]. However, nutrition training is inconsistent in different academic courses around the world and the appropriate teaching of knowledge, competencies and skills is delivered unevenly even where it would be expected [14]. Teaching of human nutrition is also generally lacking in medical training, especially for clinical aspects [14]. Similarly, this occurs also for other important topics such as physical activity. As reported by Trilk and colleagues [54], a physician’s time spent with patients discussing lifestyle behaviors, including diet, physical activity and smoking habit, was on average less than 1 min per topic.

The urgent need to respond to the burden of NCDs on the one hand, and on the other the lack of effective tools has brought out a lifestyle medicine approach. In June 2012, the American Medical Association reached a resolution to “*… urge physicians to acquire and apply … lifestyle medicine, and offer evidence-based lifestyle medicine interventions as the first and primary mode of preventing and, when appropriate, treating chronic disease within clinical medicine*” [57]. Instead of just treating symptoms, lifestyle medicine addresses the underlying causes of NCDs [54,58], employing comprehensive lifestyle changes in nutrition, physical activity, stress management, social support and environmental exposures in order to prevent, treat and even reverse the progression of chronic diseases [59]. So, every medical faculty should be committed to trying to include this novel approach in their training programs. Ultimately, it is important to acknowledge that health sciences students will be future practicing healthcare professionals, and ideally, they will themselves serve as healthy role models to promote a healthy lifestyle in the population they serve.

There are several limitations to this study worth mentioning. Because of the cross-sectional nature of the data, it is not possible to establish a causal relationship between events or draw causal inferences. Besides, differences in language between countries and the way questionnaires have been administered (paper-based or online mode) should also be considered. Again, when questionnaires were not validated in specific countries, they were adapted by using forward–backward translation, as previously described [19]. Although we obtained high response rates in most countries, differences in response rates between study sites were observed. Finally, a recall bias could have affected our results due to the approach in data collection, but since our subjects were of young age and they were required to recall the overall pattern of their habits and behaviors, we believe that this does not pose a major risk in this study.

However, there are many strengths of this study, including large sample size of students belonging to different countries and with different cultural backgrounds. This study rounded up a specific population and it attempts to make comparisons of lifestyle habits in adolescents and young adults from different cultures. It is unique, since no others before, to our knowledge, focused on such broad healthy and unhealthy behaviors among university students, considering cultural profile and inter-country differences. Particularly, the results of our study will add new knowledge and provide critical observations highlighting the need to develop tailored strategies in order to increase society‘s awareness of healthy lifestyles for NCD prevention. The results will also highlight the key role of lifestyle medicine, a new discipline that has recently emerged as a systematic approach for NCD management and prevention. Moreover, the HOLISTic study could fill the existing gap of largely unknown features and characteristics of ON [60]. There is limited information about the prevalence of ON in the research and specifically in young adults from different countries.

## 5. Conclusions

Our results reveal the presence of unhealthy lifestyle and health-related behaviors in university students, including health sciences students involved in the HOLISTic study. The current paper describes the HOLISTic study protocol and preliminary results, which is to the best of our knowledge the largest European cohort of university students surveyed for an elaborate palette of lifestyle habits. Our study provides insights in dietary patterns, sleeping habits, physical activity and perceived stress among university students, including students of health sciences, from Mediterranean and non-Mediterranean countries. We anticipate that our upcoming results will shed further light on the lifestyle habits of these students and contribute to the development of tailored interventions and strategies to be translated in lifestyle education for university students. Moreover, it is of paramount importance for healthcare students to be adequately educated for their future role of health professionals, acquiring knowledge, confidence and competence during their study program in order to deliver appropriate health interventions and healthy lifestyle advice regarding chronic diseases to their future patients.

## Figures and Tables

**Table 1 nutrients-13-00675-t001:** Overall lifestyle characteristics of students according to the study site.

	Croatia *n* = 1402	Foggia, Italy *n* = 2324	Pavia, Italy *n* = 898	Lebanon *n* = 401	Poland *n* = 245	Romania *n* = 209	Spain *n* = 535	Turkey *n* = 208	*p* ^¶^
Age (years) median (IQR)	21.0 (3.0)	23.0 (1.0)	21.0 (2.0)	19.0 (2.0)	22.0 (2.0)	24.0 (2.0)	21.0 (3.0)	21.0 (2.0)	<0.001 ^§^
Female gender *n* (%)	1096 (78.2)	1308 (56.3)	62 (69.7)	261 (65.1)	202 (82.4)	155 (74.2)	213 (39.8)	129 (62.6)	<0.001 *
Grade point average (percentile) median (IQR)	0.67 (0.2)	0.67 (0.3)	0.69 (0.3)	0.57 (0.3)	0.75 (0.2)	0.68 (0.2)	0.56 (0.2)	0.43 (0.3)	<0.001 ^§^
Smoking *n* (%)									<0.001 *
yes	323 (23.1)	590 (25.4)	192 (21.6)	105 (26.2)	61 (24.9)	74 (35.4)	64 (12.0)	45 (21.8)	
ex-smokers	189 (13.5)	134 (5.8)	55 (6.2)	35 (8.7)	36 (14.7)	28 (13.4)	62 (11.6)	17 (8.3)	
never smoked	885 (63.4)	1600 (68.8)	641 (72.2)	261 (65.1)	148 (60.4)	107 (51.2)	409 (76.4)	144 (69.9)	
BMI (kg/m^2^) median (IQR)	21.7 (3.5)	23.0 (2.4)	21.1 (3.6)	23.2 (5.8)	21.3 (4.1)	22.1 (4.9)	22.3 (2.9)	21.7 (4.3)	<0.001 ^§^
Self-rated health perception median (IQR)	9.0 (1.0)	8.0 (2.0)	8.0 (2.0)	8.0 (2.0)	8.0 (1.5)	8.0 (2.0)	8.0 (1.0)	8.0 (2.0)	<0.001 ^§^
Breakfast daily *n* (%)	765 (54.8)	1760 (75.7)	675 (75.3)	173 (43.3)	157 (64.1)	88 (42.1)	337 (64.3)	72 (36.7)	<0.001 *
ORTO-15 score median (IQR)	36.0 (5.0)	35.0 (6.0)	37.0 (6.0)	37.0 (6.0)	38.0 (8.0)	36.0 (6.0)	-	36.0 (7.0)	<0.001 ^§^
MDSS score median (IQR)	6.0 (5.0)	8.5 (6.8)	10.0 (6.0)	6.0 (5.0)	6.0 (5.0)	8.0 (6.0)	10.0 (9.0)	5.0 (6.0)	<0.001 ^§^
Sleep duration on working days (h) median (IQR)	7.0 (1.8)	8.0 (1.5)	7.7 (1.5)	7.0 (2.0)	8.0 (1.3)	7.5 (1.5)	8.0 (1.5)	7.0 (1.5)	<0.001 ^§^
Sleep duration on non-working days (h) median (IQR)	9.0 (1.5)	8.0 (2.0)	8.0 (1.5)	9.0 (2.0)	9.0 (2.0)	9.0 (2.0)	8.5 (1.0)	8.5 (1.0)	<0.001 ^§^
Total METs per week ^†^ median (IQR)	2796.0 (2964.0)	3462.0 (3885.0)	3222.0 (3480.0)	2220.0 (3501.0)	2457.0 (3133.5)	3274.5 (3740.9)	4072.8 (3829.8)	1653.0 (2572.0)	<0.001 ^§^
Daily mobile use (h) median (IQR)	3.0 (3.0)	4.0 (4.0)	3.0 (2.0)	5.0 (4.0)	3.0 (3.0)	4.0 (3.0)	3.0 (3.0)	3.0 (2.0)	<0.001 ^§^
Daily study time (h) median (IQR)	3.0 (3.0)	5.0 (3.0)	4.0 (3.0)	3.0 (2.0)	2.0 (2.0)	2.0 (3.0)	2.0 (2.0)	3.0 (3.0)	<0.001 ^§^
PSS-10 score median (IQR)	19.0 (9.0)	20.0 (7.0)	19.0 (9.0)	21.0 (7.0)	20.0 (9.0)	20.0 (6.0)	17.0 (9.0)	22.0 (7.0)	<0.001 ^§^
Students asking for more education on lifestyle and health *n* (%)	432 ^#^ (49.3)	939 (40.4)	332 (37.3)	196 (48.9)	-	65 (31.1)	261 (48.8)	57 (28.5)	<0.001 *

Legend. IQR—interquartile range, BMI—body mass index, ORTO-15—orthorexia nervosa, MDSS—Mediterranean Diet Serving Score, MET—metabolic equivalent of the task, PSS-10—Perceived Stress Scale; * chi-square test; ^§^ Kruskal–Wallis test; ^¶^ Post-hoc *p* values are presented in Appendix A; ^†^ this analysis included 4935 subjects due to missing values in International Physical Activity Questionnaire (IPAQ), ^#^ a total sample size was 877 responses in Croatia on this question (collected only in 2018).

**Table 2 nutrients-13-00675-t002:** Correlation between lifestyle habits, self-perceived health, BMI and age in the overall sample (*n* = 6222); data are presented as Spearman’s correlation coefficients and *p* values.

	Breakfast Frequency	ORTO-15	Working Days Sleep Duration	Non-Working Days Sleep Duration	Total METs Per Week	Mobile Phone Use	Study Time	PSS-10 Score	Self-Rated Health Perception	BMI	Age
MDSS score	0.22 **<0.001**	−0.15 **<0.001**	0.10 **<0.001**	−0.10 **<0.001**	0.16 **<0.001**	−0.02 0.133	0.13 **<0.001**	−0.02 0.122	0.03 **0.008**	−0.01 0.950	0.10 **<0.001**
Breakfast frequency		−0.07 **<0.001**	0.13 **<0.001**	−0.12 **<0.001**	0.04 **0.001**	−0.06 **<0.001**	0.12 **<0.001**	−0.08 **<0.001**	0.06 **<0.001**	−0.03 **0.009**	0.04 **0.006**
ORTO-15			−0.07 **<0.001**	0.06 **<0.001**	−0.11 **<0.001**	−0.04 **0.011**	−0.05 **<0.001**	−0.03 **0.030**	−0.02 0.153	−0.09 **<0.001**	−0.05 **0.001**
Working days sleep duration				0.18 **<0.001**	0.03 **0.017**	0.02 0.246	0.04 **0.001**	−0.04 **0.001**	0.03 **0.028**	−0.01 0.920	0.06 **<0.001**
Non-working days sleep duration					−0.06 **<0.001**	−0.01 0.711	−0.12 **<0.001**	0.02 0.140	−0.00 0.849	−0.05 **<0.001**	−0.08 **<0.001**
Total METs per week						−0.05 **0.001**	−0.06 **<0.001**	−0.12 **<0.001**	0.13 **<0.001**	0.09 **<0.001**	0.06 **<0.001**
Mobile phone use							0.09 **<0.001**	0.15 **<0.001**	−0.03 0.058	0.04 **0.003**	−0.06 **<0.001**
Study time								0.06 **<0.001**	0.01 0.491	0.00 0.847	0.09 **<0.001**
PSS-10 score									−0.27 **<0.001**	0.00 0.972	−0.06 **<0.001**
Self-rated health perception										−0.05 **<0.001**	0.05 **<0.001**
BMI											0.14 **<0.001**

Legend. MDSS—Mediterranean Diet Serving Score, ORTO-15—orthorexia nervosa, MET—metabolic equivalent of the task, PSS-10—Perceived Stress Scale, BMI—body mass index; *p* values < 0.05 were indicated in bold.

**Table 3 nutrients-13-00675-t003:** Association between lifestyle habits and self-rated health perception and BMI in the overall sample of students (multivariable linear regression analysis).

	Self-Rated Health Perception Beta (*p* Value)	BMI Beta (*p* Value)
Age (years)	0.01 (0.509)	0.13 (<0.001)
Female gender (males are referent)	0.03 (0.018)	0.30 (<0.001)
Smoking (nonsmokers are referent group)		
Active smokers	−0.06 (<0.001)	0.03 (0.016)
Ex-smokers	−0.00 (0.898)	−0.01 (0.347)
BMI (kg/m^2^)	−0.10 (<0.001)	na
Cohort (Croatia is referent group) *		
Foggia, Italy	−0.09 (<0.001)	0.00 (0.940)
Pavia, Italy	−0.12 (<0.001)	−0.10 (<0.001)
Lebanon	−0.08 (<0.001)	0.13 (<0.001)
Poland	−0.12 (<0.001)	−0.03 (0.009)
Romania	−0.03 (0.028)	0.02 (0.174)
Spain	−0.05 (<0.001)	−0.03 (0.053)
Turkey	−0.07 (<0.001)	−0.01 (0.264)
MDSS score	0.02 (0.171)	0.01 (0.311)
Breakfast frequency (days per week)	0.04 (0.003)	−0.01 (0.335)
Working days sleep duration (h)	0.02 (0.071)	−0.01 (0.355)
Non-working days sleep duration (h)	−0.02 (0.059)	−0.03 (0.040)
Total METs per week	0.08 (<0.001)	0.04 (0.002)
Mobile phone use (h/day)	−0.00 (0.878)	0.05 (<0.001)
Study time (h/day)	0.03 (0.039)	0.00 (0.970)
PSS-10 score (h/day)	−0.23 (<0.001)	0.04 (0.002)
Self-rated health perception	na	−0.10 (<0.001)

Legend. BMI—body mass index, MDSS—Mediterranean Diet Serving Score, MET—metabolic equivalent of the task, PSS-10—Perceived Stress Scale; na—not applicable. * Croatian students were used as a referent group based on the results obtained within the bivariate analysis of the self-rated health perception.

## Data Availability

The data presented in this study are available on request from the corresponding author. The data are not yet publicly available due to the plan for further analysis of the same dataset.

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
