# Peer review of "How Healthy Are Health-Related Behaviors in University Students: The HOLISTic Study"

_nutrients, 2021, doi:10.3390/nu13020675_

Round 1

Reviewer 1 Report

Please see comments in attached file.

Reviewer 2 Report

The paper is well constructed, structured and its results are relevant.

You should revise table 3 because some pvalues aren't in their corresponding row.

Reviewer 3 Report

This is a well-written and well-organized manuscript. The aim is interesting,  and the authors have done commendable job in highlighting and prioritizing the research questions. The abstract and tables are good. The introduction is well structured, and the discussion and conclusions well targeted.

The explanation of variables was very detailed. The discussion centered around key highlights, with suggestions for future research offered as well.

A few minor concerns:

121: The authors mention a good response rate. An overall number or a breakdown by country would provide a clearer picture. Please include the response rate in the body of the report.

221: The statistical analysis section should provide more detail, e.g. the tests based on the variable types. It is important to specify variables and variable types and the test(s) in accordance with those types

Since the analysis includes a series of variables using a multiple set of questions, it would be better to specify and explain if only completely finished survey data have been included.

There was no information about missing data provided. What percentage of data is missing and how that was handled?  Any method used to manage missing data?

In addition, how did the investigators account for recall bias, especially for dietary data analyses?

227:  Please clarify whether the regression model is multivariate or multivariable. Basically, it appears to be a multivariable regression analysis; there is no repeated measures data included in this analysis.

A post hoc analysis took place with multiple comparisons using independent and dependent statistical tests. Thus, type-1 errors are likely to occur. In order to control the Family-Wise Error Rate, the significance level needs to be determined after performing adjustments such as a Bonferroni correction. Alternatively, provide an explanation of why such adjustments are not needed.

Discussion: The study found significant differences among countries self-rated health perception and BMI, with Croatia as the reference group. Discussion includes a plausible explanation of how several factors may work together. However, the discussion should also examine how regional/country factors may have contributed to these differences, on the basis of the current findings and the previous literature.  

An explanation of the choice of Croatia as a reference group should be added.

483: Please consider adding that, because of the cross-sectional nature of the data, it is not possible to establish a causal relationship between events or draw causal inferences.

Author Response

Please, see the attachement
